Subject Areas:
nanotechnology/statistics

Keywords:
Box–Behnken design, double emulsion technique, hypertension, nanoparticles, telmisartan

Author for correspondence:
Pradipta Sarkar
e-mail: pradiptasarkar.rs@jadavpuruniversity.in

This article has been edited by the Royal Society of Chemistry, including the commissioning, peer review process and editorial aspects up to the point of acceptance.

# Application of statistical design to evaluate critical process parameters and optimize formulation technique of polymeric nanoparticles

Pradipta Sarkar[1,2], Saswati Bhattacharya[2] and Tapan Kumar Pal[1]

[1]Bioequivalence Study Centre, Jadavpur University, Kolkata-700032, India
[2]Department of Pharmaceutical Technology, Jadavpur University, Kolkata, India

PS, 0000-0002-9881-9385

In advanced medication, drug-loaded polymeric nanoparticles (NPs) appeared as a novel drug delivery system with lots of advantages over conventional medicines. Despite all the advantages, NPs do not gain popularity for manufacturing hurdles. The study focused on the formulation difficulties and implementation of statistical design to establish an effective model for manufacturing NPs. In this study, physico-chemical properties of the drug and polymer (PLGA) were incorporated to understand the mechanistic insights of nanoformulations. Primarily, the process controlling parameters were screened by Plackett–Burman design and the critical process parameters (Cpp) were further fabricated by Box–Behnken design (BBD). The TLM-PLGA-NPs (telmisartan loaded PLGA NPs) exhibited particle size, encapsulation efficiency and zeta potential of 232.4 nm, 79.21% and −9.92 mV respectively. The NPs represented drug loading of 76.31%. Korsmeyer–Peppas model ($R^2 = 0.925$) appeared to be the best fitted model for *in vitro* release kinetics of NPs. The model identified Fickian diffusion of TLM from the polymeric nanoparticles. The ANOVA results of variables indicate that BBD is a suitable model for the development of polymeric NPs. The study successfully identified and evaluated the correlation of significant parameters that were directly or indirectly influencing the formulations which deliberately produce desired nanoparticles with the help of statistical design.

# 1. Introduction

In the modern era, attribute of pharmaceutical products are described by its quality. The quality pioneer, Dr Joseph M. Juran stated that every product should have a certain quality which may be obtained by considering the crisis in quality and obstacles in the way of product development. Pharmaceutical quality by design (QbD) is a systemic approach of product development that focuses on predefined functionality of the product for understanding product and process optimization, quality risk management, reduction of variation and improvement of process efficiency. The concept of QbD is becoming very essential, linking the critical process parameter (CPP) and critical material attributes (CMA) to critical quality attributes (CQA) for building quality into the products. QbD approach is also encouraged by the various regulatory authorities to maintain uniformity of the products in meaningful quality specifications.

Statistical design is one of the major predictive measures in QbD to develop, manufacture and optimize a product by applying mathematical and statistical methodology to experimental data. Generally, statistical design is employed at the beginning, where significant factors are screened from a large number of variables. Normally, significant independent variables are selected by two-level factorial design (Plackett–Burman design—PBD) that influences responses of the development work [1]. Further, the response data are analysed to form a second-degree polynomial equation to describe the relation between independent and dependent variables of the system. Such independent variables may influence the responses by simultaneous variation in experimental model. Among the various statistical optimization techniques, Box–Behnken design (BBD) of response surface methodology (RSM) is considered the most promising technique for analysing the multivariate in the process or product development [2–4]. It is an efficient and widely used methodology under RSM to optimize the nanoformulation with a limited number of experiments [5,6].

Some essential medicines like telmisartan (TLM), which became a first line of treatment, are generally prescribed once daily (*quaque die*) to control hypertension throughout the day. Importance of the antihypertensive medication can be imagined from the reported data published by Nwankwo T *et al.* in NCHS Data Brief 2013 [7]. Futuristic projection about hypertension suggests the higher end value of blood pressure of 1.56 billion individuals who are likely to be suffering from hypertension by 2025 [8]. It is the major risk factor for patients suffering from cardiovascular diseases, accounting for 9.4 million deaths each year [9].

TLM, an Angiotensin II receptor blocker (ARB) [10] is well known for its tolerability profile [11] in the treatment of essential hypertension. It is a highly selective AT1 receptor antagonist and one of the first ARBs of choice at high-risk vascular events [12,13]. Among all ARBs, this is the first drug which received approval from the US and European medical agencies [14].

In this study, TLM was introduced as a model drug to make some advancement on the dosage form and manufacturing process which can be consumed on a daily basis by a large number of patients. In this regard, the chemical nature of TLM was studied for better understanding of manufacturing process with statistical optimization. Consideration of physico-chemical aspect of drug moiety (TLM) with manufacturing process has led to the study being more interpretable. TLM, chemically described as 4′-[(1,4′-dimethyl–2′-propyl[2,6′-bi-1H-benzimidazol] 1′-yl)methyl]-[1,1′-biphenyl]-2-carboxylic acid is a nonpeptide amphiphilic molecule having benzimidazole structure with heterocyclic substituents [15]. The solubility characteristic of TLM in water and organic solvent [16] plays a vital role in emulsification process. Lipophilic surface environment of TLM demonstrates a high partition coefficient (log $p = 3.2$) [14] which decreases solubility into water. However TLM shows good solubility at low and high pH (<3; >8). Few organic solvents can solubilize TLM in a limited extent. The chemical property and intermolecular energy lead TLM to confine the crystalline structure as well [17]. The distinct physico-chemical characteristics of TLM [16] also regulate biopharmaceutical property of the moiety. It is already known that crystalline structure has low soluble and poor biopharmaceutical property rather than amorphous TLM [18]. Therefore, several attempts were made to improve biopharmaceutical property of TLM by converting crystal to amorphous moiety [19]. Although in conventional dosage form, TLM exhibited wide range of bioavailability in different subjects [20,21].

The study demonstrates the statistical optimization process accounting physico-chemical property of drug to establish high drug-encapsulated uniform nanoparticles (NPs). Entrapment into polymer matrix of this deceptive molecule was extensively studied and characterized for better understanding of the design space. The statistical models applied in development studies were not discussed deeply because it has been well discussed in different scientific literature [1,22,23]. The main intention lies between the statistical models and physico-chemical property of drug molecule in formulation process. Proper application of statistical models can also reduce production cost of expensive polymeric nanoparticles. The developed

formulation can be applied in many fields including drug targeting [24], sustained release treatment and implants [25]. The study aimed to identify varied critical process attributes in the way of developing high-quality product and optimizing the parameters focusing on reliable clinical application of drug product.

# 2. Material and methods

## 2.1. Materials: (chemicals, reagents and application of instruments)

Poly(D,L-lactide-co-glycolide) (Resomer® RG504H) and Poly (vinyl alcohol) (MW 31 000–50 000, 98–99% hydrolysed) were purchased from Sigma Aldrich Chemicals Pvt Limited, Karnataka 560100, India. TLM was received as a gift sample from Aurobindo Pharma, Hyderabad, India. Telpres 80 (TLM 80 mg) tablets were obtained from Abbott Laboratories, Mumbai, India. Dichloromethane (DCM) and Polysorbate 80 (Tween 80) were purchased from Merck India. Ethanol was obtained from Merck KGaA, Germany. Deionized water was provided by Bioequivalence Study Centre, Depart. of Pharm. Tech., Jadavpur University, Kolkata, India. IKA T10 basic Turrax Ultra homogenizer was used to formulate nanoemulsion. Other instrument facilities like liquid chromatography (LC) and mass spectrometer (MS) (API2000, AB Sciex), a high-speed cold centrifuge and lyophilizer were provided by Bioequivalence Study Centre of the same department.

## 2.2. Preparation of nanoparticles

A modified w/o/w-emulsion solvent evaporation technique was employed to prepare polymeric NPs. Briefly, PLGA and drug were dissolved in 2 ml of an organic solvent (DCM-ethanol) system which was developed during the formulation. Initially, solubility characteristic of TLM was studied by LC-MS/MS. The developed solvent system was further used for determining the solubility of PLGA. The primary ($w_1$) and secondary water ($w_2$) phase of the emulsion were prepared by dissolving PVA (polyvinyl alcohol) (emulsifier) in deionized water in different ratios (table 3). The primary water phase ($w_1$) was emulsified with the oil phase using high-speed homogenization (25 000 r.p.m.) for 5 min. Then the primary emulsion ($w_1/o$) was introduced into secondary water phase ($w_2$) containing 0.5% PVA by a syringe, maintaining the homogenization speed at 7000 r.p.m.. Additionally the prepared w/o/w emulsion was sonicated at 20°C for 30 min. After that, the emulsion was allowed to stay overnight (10 h) on the magnetic stirrer (at 320 r.p.m.) to evaporate the organic solvents. The NPs were collected by centrifugation method at 16 000 r.p.m. for 30 min at 4°C. Washing was done three or four times with deionized water until complete removal of the surfactant (PVA) and finally the NPs were collected by centrifugation for 10 min at the same speed and temperature as given above. The collected NPs were lyophilized to dry completely and stored at −20°C for future work.

## 2.3. Experimental design

The entire manufacturing process was fabricated using various statistical predictive models. At the beginning, all feasible variables were screened by PBD (table 1) to obtain three most effective independent variables. Independent variables (factors) including the PLGA content ($X_1$), percentage of polyvinyl alcohol ($X_2$) as surfactant in the primary emulsion ($w_1/o$) and homogenization speed ($X_3$) introduced with BBD to generate 17 predictive runs. The independent variables were characterized in three different levels and coded as −1, 0 and +1 for low, medium and high level respectively as mentioned in table 2. The desired responses were optimized by using BBD to construct polynomial equations. Responses (dependent variables) of the corresponding factors were selected as particle size, encapsulation efficiency (%EE) and zeta potential as described in table 2. This design allowed estimation of the parameters through second-order polynomial equations obtained by Design-Expert software (Version 11, Stat-Ease Inc., Minneapolis, MN, USA). Based on this software, a total of 17 experimental runs were conducted to develop the mathematical models as shown in table 3. Finally the quadratic model was expressed by second-order polynomial equation (equation (2.1)) as given below.

$$Y = \beta_0 + \beta_1 X_1 + \beta_2 X_2 + \beta_3 X_3 + \beta_4 X_1 X_2 + \beta_5 X_2 X_3 + \beta_6 X_1 X_3 + \beta_7 X_1^2 + \beta_8 X_2^2 + \beta_9 X_3^2, \qquad (2.1)$$

where

$Y$ = predicted response,
$\beta_0$ = intercept,
$\beta_1$, $\beta_2$ and $\beta_3$ = linear coefficients,
$\beta_4$, $\beta_5$ and $\beta_6$ = interaction coefficients,
$\beta_7$, $\beta_8$ and $\beta_9$ = square coefficients.

**Table 1.** Experimental matrix of PBD. Note: Hz—homogenization, Mw—molecular weight, T-group—terminal group, adj—adjusted, pred—predicted, adeq—adequate.

**response 1: particle size**

| source | sum of squares | mean | $F$-value | $p$-value | fit statistics | | |
|---|---|---|---|---|---|---|---|
| model | 21 614.26 | 2701.78 | 183.25 | 0.000603 | std. dev. | 3.83 | $R^2$ | 0.99 |
| PLGA cont. | 845.04 | 845.04 | 57.31 | 0.00478 | mean | 276.45 | adj $R^2$ | 0.99 |
| PVA% | 212.52 | 212.52 | 14.41 | 0.032076 | C.V.% | 1.38 | pred $R^2$ | 0.96 |
| Hz speed (r.p.m.) | 13 220.24 | 13 220.24 | 896.70 | $8.18 \times 10^{-5}$ | | | adeq. precision | 38.71 |
| Hz duration | 95.76 | 95.76 | 6.49 | 0.08404 | | | | |
| PVA Mw | 4989.84 | 4989.84 | 338.45 | 0.00035 | | | | |
| PLGA T-group | 75.50 | 75.50 | 5.12 | 0.108626 | | | | |
| residual | 44.22 | 14.74 | | | | | | |
| cor total | 21 658.49 | | | | | | | |

**response 2: encapsulation efficiency**

| source | sum of squares | mean | $F$-value | $p$-value | fit statistics | | |
|---|---|---|---|---|---|---|---|
| model | 570.71 | 71.33 | 63.88 | 0.0028 | std. dev. | 1.056 | $R^2$ | 0.99 |
| PLGA cont. | 367.41 | 367.41 | 329.026 | 0.00036 | mean | 71.16 | adj $R^2$ | 0.97 |
| PVA% | 111.63 | 111.63 | 99.96 | 0.0021 | C.V.% | 1.48 | pred $R^2$ | 0.90 |
| Hz duration | 17.28 | 17.28 | 15.47 | 0.029257 | | | adeq. precision | 24.33 |
| PVA Mw | 17.28 | 17.28 | 15.47 | 0.029257 | | | | |
| PLGA T-group | 6.16 | 6.16 | 5.51 | 0.1003 | | | | |
| PLGA : drug ratio | 8.003 | 8.003 | 7.16 | 0.075 | | | | |
| residual | 3.35 | 1.11 | | | | | | |
| cor total | 574.06 | | | | | | | |

(*Continued.*)

**Table 1.** (*Continued.*)

| response 3: zeta potential | | | | | fit statistics | | | |
|---|---|---|---|---|---|---|---|---|
| source | sum of squares | mean | F-value | p-value | | | | |
| model | 141.12 | 17.64 | 593.49 | 0.0001 | std. dev | 0.17 | $R^2$ | 0.99 |
| PLGA cont. | 1.68 | 1.6875 | 56.7757 | 0.0048 | mean | −6.70 | adj $R^2$ | 0.99 |
| PVA% | 2.16 | 2.1675 | 72.92 | 0.003 | CV.% | 2.56 | pred $R^2$ | 0.98 |
| Hz speed (r.p.m.) | 1.54 | 1.54 | 51.84 | 0.005 | adeq. precision | | | 62.28 |
| PLGA Mw (kDa) | 3.74 | 3.74 | 125.85 | 0.0015 | | | | |
| PLGA T-group (kDa) | 128.70 | 128.70 | 4330.34 | $7.7 \times 10^{-6}$ | | | | |
| PLGA : drug ratio | 1.40 | 1.40 | 47.13 | 0.0063 | | | | |
| residual | 0.089 | 0.029 | | | | | | |
| cor total | 141.20 | | | | | | | |

**Table 2.** Variables and their levels studied in BBD.

| independent variables | | levels | | |
| --- | --- | --- | --- | --- |
| | | −1 | 0 | +1 |
| $X_1$ | polymer cont. (mg) | 100 | 150 | 200 |
| $X_2$ | PVA (%) | 0.5% | 2.75 | 5% |
| $X_3$ | homogenization speed (r.p.m.) | 10 000 | 20 000 | 30 000 |
| dependent variables | | constraints | | |
| $Y_1$ | particle size (nm) | minimize | | |
| $Y_2$ | encapsulation efficiency (%) | maximize | | |
| $Y_3$ | zeta potential (mV) | maximize | | |

**Table 3.** BBD of three-variable systems.

| run (NF) | independent variables | | | dependable variables | | |
| --- | --- | --- | --- | --- | --- | --- |
| | polymer cont. | % PVA | homogenization speed | particle size (nm) | encapsulation efficiency (%) | zeta potential (mV) |
| 1 | 1 | 0 | 1 | 243.71 | 71.15 | −10.66 |
| 2 | 1 | 0 | −1 | 317.29 | 75.23 | −10.17 |
| 3 | −1 | −1 | 0 | 252.51 | 35.41 | −4.32 |
| 4 | 0 | −1 | −1 | 293.63 | 60.38 | −8.51 |
| 5 | 0 | 1 | −1 | 282.28 | 65.6 | −6.06 |
| 6 | 0 | 0 | 0 | 228.64 | 69.37 | −9.29 |
| 7 | 1 | 1 | 0 | 275.21 | 81.29 | −7.11 |
| 8 | −1 | 0 | −1 | 261.35 | 36.79 | −5.74 |
| 9 | 0 | 0 | 0 | 221.3 | 74.15 | −9.92 |
| 10 | 0 | 0 | 0 | 228.41 | 72.63 | −9.66 |
| 11 | 0 | 0 | 0 | 231.17 | 70.34 | −9.14 |
| 12 | 0 | 0 | 0 | 223.47 | 72.16 | −9.97 |
| 13 | −1 | 0 | 1 | 202.25 | 38.21 | −5.83 |
| 14 | 1 | −1 | 0 | 285.72 | 68.16 | −10.73 |
| 15 | −1 | 1 | 0 | 215.38 | 36.15 | −4.86 |
| 16 | 0 | 1 | 1 | 205.47 | 61.27 | −8.19 |
| 17 | 0 | −1 | 1 | 225.38 | 60.07 | −9.02 |

## 2.4. Optimization and validation of the design

The desirability function, which is a geometric mean of all transformed responses is the most important and most currently used multi-criteria methodology used for optimization. In the concerned BBD, the desirability function approach was applied to create best-fitted values of operating variables to obtain a desirable response in compliance with the selected criteria. Based on the desired output of particle size (minimum), encapsulation efficiency (maximum) and zeta potential (maximum) of the nanoformulation, a suggestion was provided by the software along with the desirability score ranged between 0 to 1. In this context, a desirability score towards 1 indicates about the predicted values of desired response. As directed, the combinations were used to prepare an optimized formulation. The values obtained from predicted and experimental formulations were compared and the error percentages were evaluated [26].

## 2.5. Characterization of polymeric nanoparticles

### 2.5.1. Fourier transform infrared (FTIR) analysis

Chemical integrity between PLGA, drug and other excipient (PVA) of the NPs was determined by identifying specific bond vibration of each ingredient using FTIR spectroscopy (model: Nicolet iS10, Thermo Fisher Scientific, USA). The samples were scanned in the wavenumber range from 400 to $4000\,cm^{-1}$.

### 2.5.2. Differential scanning calorimetry (DSC) and X-ray diffraction (XRD) analysis

The structure alteration of materials due to heat exchange was measured in order to determine the integrity of the NPs. DSC (Perkin Elmer Pyris Diamond) was used to measure heat exchange capacity in a controlled inert environment at heat rate $10°C\,min^{-1}$ in a temperature range of 5°C to 350°C under constant $N_2$ flow of $40\,ml\,min^{-1}$ [27].

X-ray diffraction of excipients indicates crystallinity which helps to identify the physical state of the compounds. The physical state of pure drug, PLGA and NPs was evaluated using Ultima III (Rigaku) X-ray diffractometer (XRD) with copper target and K-beta filter. Powder sample was mounted on a quartz plate and smoothed to a level surface. The XRD pattern of each sample was measured over an angular range from 5° to 90° (diffraction angle $2\theta$) in stepwise increments of 0.05° and a scanning speed of $3°\,min^{-1}$ radiation.

### 2.5.3. Liquid chromatography and mass spectrometer (LC-MS/MS) analysis

Pure TLM was dissolved in methanol (HPLC grade) and was infused in mass spectrometer for Q1 scan. Extracted TLM from NPs was also injected into LC-MS/MS to compare with pure drug. Quantification of TLM in NPs was determined by LC-MS/MS to evaluate encapsulation efficiency of NPs, drug loading and *in vitro* drug release kinetics.

### 2.5.4. Surface morphology of nanoparticles

The physical structure and surface morphology of the NPs were observed with the help of scanning electron microscopy (SEM, EVO 18, Special Edition, Zeiss). The NPs were mounted on an aluminium dais using carbon adhesive. A thin film of gold was spotted over the samples in high vacuum under low pressure using a Quorum Q150T ES.

### 2.5.5. Determination of particle size and zeta potential

The physical dimension (particle size), surface potential (zeta potential) and size distribution of NPs were measured by dynamic light scattering technique using a Zetasizer (Nano-ZS90, Malvern Instruments, UK). A small amount (0.1%) of samples (NPs) was dispersed properly in deionized water and poured into a cuvette for analysis [28].

### 2.5.6. Determination of encapsulation efficiency (%EE) and percentage of drug loading (%DL)

The quantification of drug into NPs was performed using a validated method in LC-MS/MS. The encapsulated drug was separated from the polymer (PLGA) matrix by the following method and introduced in LC-MS/MS to determine drug content of NPs. NPs (1 mg) were dissolved in DCM to extract the encapsulated TLM. Dissolved PLGA was separated by centrifugation at 12 000 r.p.m. after precipitation with 2 ml of methanol. The supernatant solution was collected and dried under $N_2$ atmosphere at 45°C. The dried sample was reconstituted with a solvent mixture of methanol : water in 1 : 1 ratio. The reconstituted samples were poured into auto-sampler vial for quantification. Entrapment efficiency (%) and drug loading (%) were calculated using equations (2.2) and (2.3) respectively as described earlier [29].

$$\text{encapsulation efficiency (\%EE)} = \frac{\text{weight of drug in nanoparticles}}{\text{initial weight of feeding drug}} \times 100 \qquad (2.2)$$

and

$$\text{drug loading (\%DL)} \quad = \frac{\text{weight of drug in nanoparticles}}{\text{weight of nanoparticles}} \times 100. \qquad (2.3)$$

## 2.6. *In vitro* drug release study

Membrane diffusion technique was employed to evaluate *in vitro* release profile of TLM-loaded NPs [30]. Briefly, the optimized formulation (2 mg) was suspended into 4 ml of phosphate buffer solution (pH 7.4) containing 1% Tween 80 solution. The suspended NPs were sealed into a dialysis sac (i.d. 14.3 mm, MW cut-off, 12 000–14 000, HiMedia, India) which was then placed into the reservoir containing 100 ml of above-mentioned solution. Since TLM is insoluble in water, solubilizer (1% of Tween 80) was mixed with the medium. The medium was stirred (100 r.p.m.) continuously at controlled temperature of 37° C. The solution (1 ml) was withdrawn and replaced with the same volume of fresh solution at predetermined time interval [31]. The dissolution study of TLM tablet (80 mg) was also performed simultaneously according to the method described by Gaur *et al.* [32]. Thereafter collected samples were extracted and introduced in LC-MS/MS for quantification. The cumulative percentage of drug into the medium was calculated from the LC-MS/MS data to plot a curve with function of time.

For statistical analysis, principally practised model-dependent approach was selected to evaluate the release profile of TLM from polymeric NPs, i.e. zero-order model [33], first-order model [33], Higuchi model [34] and Korsmeyer–Peppas model [35]. The result obtained from *in vitro* drug release study was treated for complementing the above four models. The highest coefficient of determination ($R^2$) was considered as a determinant factor for describing the most suitable kinetic model of TLM-loaded NPs [34].

# 3. Results and discussion

## 3.1. Evaluation of manufacturing process of nanoparticles

In the present study, a solubility modulated double emulsion (w/o/w) solvent evaporation technique was employed to prepare uniform, spherical and drug-loaded polymeric NPs. The double emulsification technique gave the opportunity to incorporate two different compartments into single nanodroplet. The appropriate concentration of primary emulsifier (PVA) made the w/o emulsion stable enough to bring uniformity in particle size, which reflected in secondary emulsion (w/o/w) [36,37]. The minimum mechanical stress (320 r.p.m.) ensured the mobility during transformation of nanodroplet to NPs. The polymeric NPs were developed by focusing on simplified formulation technique considering the minimum application of the excipients. The selection of each variable was accomplished by following the methodology described in ICH (Q11) guideline [38]. At the beginning of development and assessments of product attributes, necessary knowledge regarding potential variables, appropriate methodology and control strategy was gathered from previously published literature to achieve a formulation with all desired characteristics [39,40]. Each of the possible product or process variables was incorporated in PBD (electronic supplementary material, table S2) for identification of major affecting parameters on the possible outcome of the desired responses [41,42]. Among all independent variables (electronic supplementary material, table S1), PLGA content, percentage of PVA and homogenization speed were identified as major controlling parameters because of large *p*-value calculated by PBD (table 1). The PBD analysis provided a good scientific rationale about the major process affecting parameters that need to be controlled (electronic supplementary material, figures S1–S3). Based on the criticality, the highest influencing attributes were controlled, to ensure desired output and process validation [43]. The key independent parameters were maintained in a narrow range to validate consistency in process performance and robustness. The three different levels (listed in table 2) of selected variables made the study versatile to predict centre points (5 points) of BBD. A scientifically justified model was developed that can account for prediction of quality, and support extrapolation of operating conditions for diverse applications of polymeric nanoparticles in its current form. The process controlling parameters were further discussed in their individual section of the article.

Formulation number (NF) 1, 2 and 7 (table 3) clearly indicates that droplet size is inversely proportional to homogenization speed and time [44]. During this period nanodroplets were formed in the emulsion (w/o) by shear force of the turbulence generated in between the gap of rotor and stator

[44–46]. Therefore, viscosity of both dispersed and continuous phase played an important role to reduce size and stabilize the droplets. Consequently, equilibrium in droplet size of nanoemulsion was not achieved due to high viscous dispersed phase (NF 5). More shear stress (homogenization speed) was applied in highly viscous medium to break down the dispersed phase into small droplets (NF 16). However, excessive mechanical energy could also produce small droplets which had coalescence in an unstable colloidal system (NF 3). On the other hand, the high viscosity of the continuous phase helps to get maximum encapsulation efficiency by stabilizing the small droplets (NF 7).

The significant range of each independent variable was established by performing initial screening, which was presented in table 2. Results (table 4) indicate BBD was found as a suitable statistical design for the development work in a limited number of experimental runs [3,26]. The model suggested 17 experimental runs with five centre points (table 3). The polynomial equations were established with significant coefficient obtained by analysing the observed value using ANOVA and which has been mentioned in table 4. The combined influences of the factors e.g. $X_1X_2$, $X_2X_3$ and $X_1X_3$ could be revealed from the quadratic equations of each response. Significant square values of independent variables were the identity of the quadratic model. The study also evaluates the correlation of significant parameters that are directly or indirectly influencing the formulation during statistical optimization.

## 3.2. Determination of particle size of nanoparticles

The mean diameter of the NPs, ranging from 202.25 to 317.29 nm has been found in 17 suggested runs (table 3) of the BBD. The polynomial equation (equation (3.1)) exhibited direct effect on the particle size of the independent variables (i.e. $X_1$, $X_2$, $X_3$). The positive sign indicated that $X_1$ has a favourable effect and negative sign for both $X_2$ and $X_3$ revealed the reverse influence on the particle size. Analysis of data from ANOVA test exhibited the rise in average particle diameter with an increase in PLGA content. The high polymer (PLGA) concentration possesses increased viscosity (NF 1) of the PLGA solution which required more shear stress to break into small pieces [45]. Therefore, a limited homogenization speed (NF 2) could not deliver sufficient shear stress which usually leads to large particle diameter (figure 1c) [47]. The droplet in primary emulsion acted like an indicator of particle size which then diffused in a large volume water phase in the secondary emulsification stage. In this phenomenon, surfactant also played an important role in stabilization by preventing coalescence of the nanoemulsion. In this way the model revealed the small size of NPs obtained with the increased value of PVA concentration and homogenization speed (NF 16), in a similar way as illustrated by Lee *et al.* [48] (figure 1e). The three-dimensional representation of response surface plot in figure 1 also demonstrates the variation of particle size due to changes in the independent variables. The model demonstrates a predicted $R^2$ value of 0.9460 which is in reasonable agreement with the adjusted $R^2$ value of 0.9848. The model with a higher adequate ratio of 36.66 expresses suitable signal strength which may be used to navigate the design space. $F$-value of 116.21 implies that the model is significant for predicting desired particle diameter of NPs. The polynomial regression equation relating to particle size and independent variables was constructed with significant coefficients ($p < 0.05$) as shown below.

$$Y_1 = 226.60 + 23.81X_1 - 9.86X_2 - 34.72X_3 + 6.65X_1X_2 + 17.53X_{12} + 13.07X_{22} + 12.02X_{32}. \tag{3.1}$$

## 3.3. Entrapment and loading efficiency measurement

The encapsulation efficiency of various suggested runs was varied from 35.41% to 81.29% with different combination of factors and levels (table 3). The model $F$-value of 106.97 indicates that the second order response surface model is significant at the 5% level. The predicted $R^2$ value and adjusted $R^2$ value were found to be 0.9397 and 0.9835, respectively, that are also in a reasonable order. An adequate ratio of 28.84 demonstrates a satisfactory signal which indicates that the model can be used in navigating design space. The polynomial equation (equation (3.2)) describes that the independent factors like $X_1$ and $X_2$ have a significant effect on drug encapsulation (NF 1, 8). This had also been projected in the figure 2a,b. The lipophilic character of TLM [20] is an advantage to entrap into PLGA which minimizes drug diffusion. But the crystalline nature of TLM reduced the entrapment efficiency of NPs. Amorphous nature of drug into NPs (figure 7) also indicates that the drug may coexist across the polymer matrix, which leads to high encapsulation efficiency. The high PVA concentration reduces the surface tension of primary emulsion (w/o) leading to drug diffusion into the core of the particles (NF 5). The positive coefficient and three-dimensional plot of $X_1$ and $X_2$ factors also support the phenomenon i.e. the

**Table 4.** ANOVA results in BBD for particle size, EE and zeta potential.

| source | particle size | | | | | encapsulation efficiency | | | | | zeta potential | | | | |
| --- | --- | --- | --- | --- | --- | --- | --- | --- | --- | --- | --- | --- | --- | --- | --- |
| | sum of | d.f. | mean | F-value | p-value | sum of | d.f. | mean | F-value | p-value | sum of | d.f. | mean | F-value | p-value |
| model | 18 120.56 | 9 | 2013.40 | 116.21 | 0.00 | 3724.17 | 9 | 413.80 | 106.97 | 0.00 | 69.91 | 9 | 7.77 | 48.08 | 0.00 |
| PLGA cont. (A) | 4533.42 | 1 | 4533.42 | 261.65 | 0.00 | 2785.19 | 1 | 2785.19 | 720.00 | 0.00 | 40.14 | 1 | 40.14 | 248.45 | 0.00 |
| PVA (B) | 778.15 | 1 | 778.15 | 44.91 | 0.00 | 51.46 | 1 | 51.46 | 13.30 | 0.01 | 5.06 | 1 | 5.06 | 31.29 | 0.00 |
| homogenization speed (C) | 9642.44 | 1 | 9642.44 | 556.53 | 0.00 | 6.66 | 1 | 6.66 | 1.72 | 0.23 | 1.30 | 1 | 1.30 | 8.02 | 0.03 |
| AB | 177.16 | 1 | 177.16 | 10.22 | 0.02 | 38.38 | 1 | 38.38 | 9.92 | 0.02 | 4.33 | 1 | 4.33 | 26.78 | 0.00 |
| AC | 52.42 | 1 | 52.42 | 3.03 | 0.13 | 7.56 | 1 | 7.56 | 1.95 | 0.20 | 0.04 | 1 | 0.04 | 0.25 | 0.63 |
| BC | 18.32 | 1 | 18.32 | 1.06 | 0.34 | 4.04 | 1 | 4.04 | 1.04 | 0.34 | 0.66 | 1 | 0.66 | 4.06 | 0.08 |
| A$\hat{A}^2$ | 1294.42 | 1 | 1294.42 | 74.71 | 0.00 | 555.03 | 1 | 555.03 | 143.48 | 0.00 | 7.59 | 1 | 7.59 | 47.00 | 0.00 |
| B$\hat{A}^2$ | 719.65 | 1 | 719.65 | 41.54 | 0.00 | 105.11 | 1 | 105.11 | 27.17 | 0.00 | 9.45 | 1 | 9.45 | 58.48 | 0.00 |
| C$\hat{A}^2$ | 608.19 | 1 | 608.19 | 35.10 | 0.00 | 101.25 | 1 | 101.25 | 26.17 | 0.00 | 0.10 | 1 | 0.10 | 0.61 | 0.46 |
| residual | 121.28 | 7 | 17.33 | | | 27.08 | 7 | 3.87 | | | 1.13 | 7 | 0.16 | | |
| lack of fit | 55.07 | 3 | 18.36 | 1.11 | 0.44 | 12.73 | 3 | 4.24 | 1.18 | 0.42 | 0.58 | 3 | 0.19 | 1.41 | 0.36 |
| pure error | 66.21 | 4 | 16.55 | | | 14.35 | 4 | 3.59 | | | 0.55 | 4 | 0.14 | | |
| cor total | 18 241.84 | 16 | | | | 3751.25 | 16 | | | | 71.04 | 16 | | | |

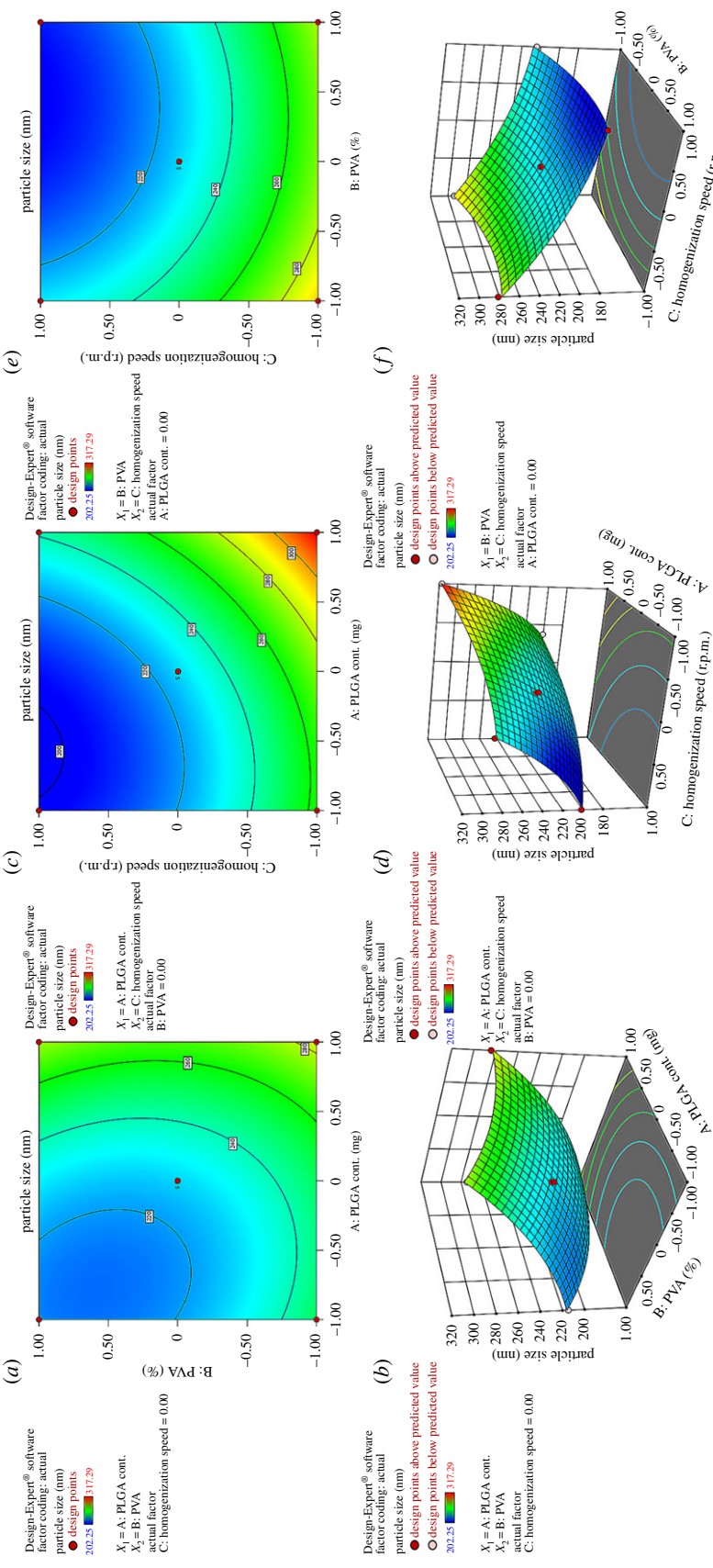

**Figure 1.** Contour plots and three-dimensional response surface plots showing the effect of different variables on the particle size of TLM-loaded NPs.

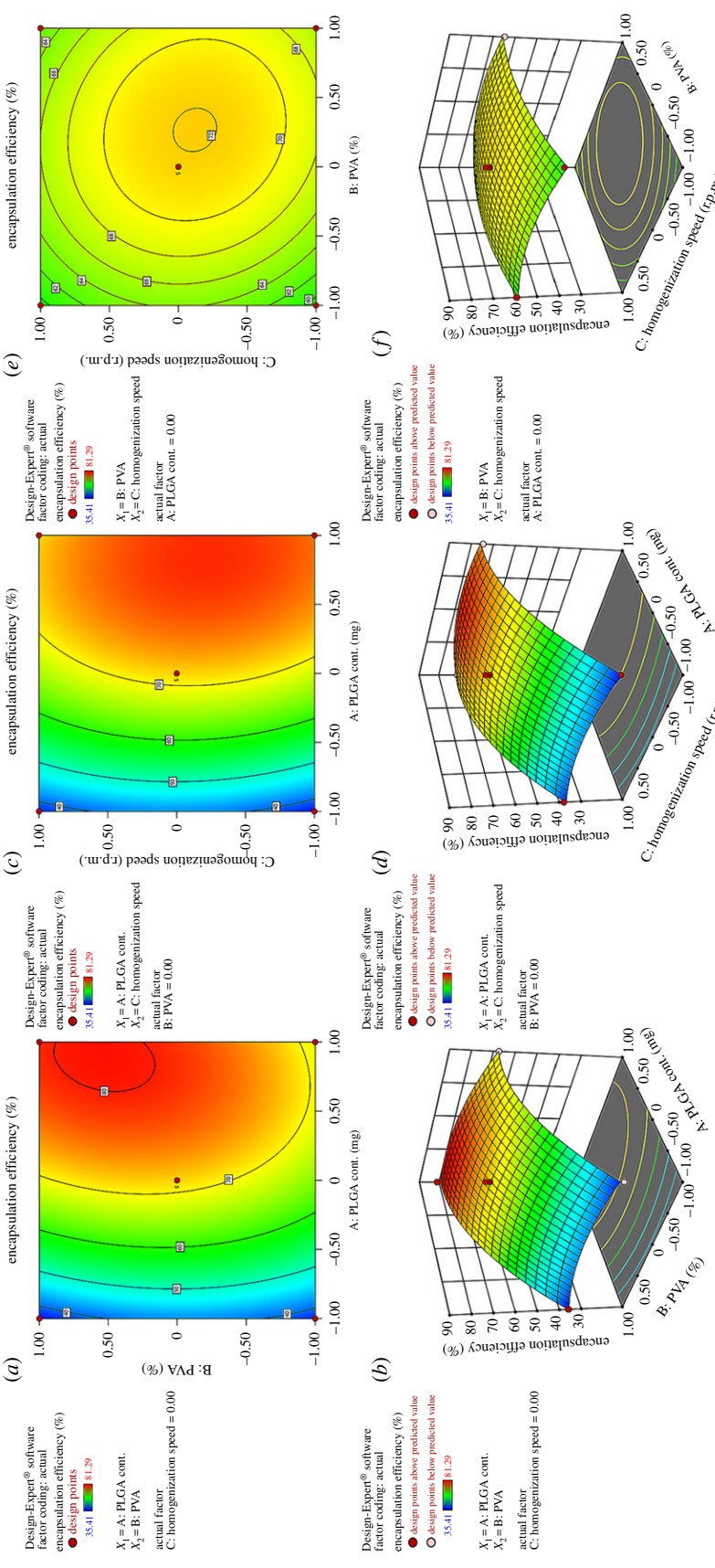

**Figure 2.** Contour plots and three-dimensional response surface plots showing the effect of different variables on the encapsulation efficiency of TLM-loaded NPs.

responses increase with the factor and vice versa. The $X_3$ has also contributed in drug encapsulation indirectly by producing uniform NPs (figure 2$c$,$d$). The model terms of the polynomial equation were considered in the basis of the values of 'Prob > F' which was less than 0.05. The model proposed a good fit polynomial equation ($R^2 = 0.99$) for encapsulation efficiency, as mentioned below.

$$Y_2 = 71.73 + 18.66X_1 + 2.54X_2 + 3.10X_1X_2 - 11.48X_{12} - 5.00X_{22} - 4.90X_{32}. \tag{3.2}$$

## 3.4. Determination of zeta potential of nanoparticles

From a viewpoint of double emulsion solvent evaporation method, zeta potential stands for a potential difference between the stationary phase of dispersed particles and the dispersion medium [49]. Its value can be related to the stabilities (short- and long-term) of colloidal dispersion where the higher value signifies a relatively stable system when the repulsive forces exceed the attraction force working among the particles [50], while at the same time emulsions with low electric potential tend to coagulate or flocculate, resulting in poor physical stability [51]. In the present study, zeta potential values were obtained in ranges between –4.32 and –10.73 mV. The model $F$-value of 48.08 implies suitability of the model with experimental data. There is a 0.01% chance that a 'model $F$-value' as large as this could appear due to noise. Predicted $R^2$ value of 0.8572 was found to have a good correlation with an adjusted $R^2$ of 0.9636. A desirable signal to noise ratio was determined by adequate precision value greater than 4. Thus an adequate value of 21.32 signifies an adequate signal to navigate the optimization process. The significant model terms were selected based on the $P > F$ value less that 0.05 (table 4). However, values greater than 0.1 indicate an insignificant term of the model, although the term below the above value was not considered here to form a quadratic equation of zeta potential. The corresponding polynomial equation (equation (3.3)) established positive relation with $X_1$ and $X_2$ and showed a negative relation with $X_3$ for zeta potential.

Dustin L. Cooper and Sam Harirforoosh stated, 'The anionic characteristics of PVA led to the formation of NPs with slightly negative surface charges' [52]. However, in this study zeta potential values were found to decrease with the increase in the concentration of PVA (figure 3$e$,$f$) and (NF 4, 5). The possible reason might be the residual PVA [53] that could have masked the charge group on the surface of the NPs. It may effectively form a shield over the NPs that can lead to lower zeta potential value [54,55].

In this study higher polymer concentration exhibited nanoformulation with increased zeta potential (figure 3$a$ and figure 2$b$) and (NF 2, 13). The polymer concentration in primary emulsion was found to be the major contributing factor to the value of zeta potential. The reason could be the negative charge that manifested in the presence of terminal carboxylic groups of PLGA copolymer on NPs [55,56].

The homogenization speed may have contributed a little influence on zeta potential (figure 3$c$,$d$). However, the values were found to increase with the increasing value of stirring speed (NF 1, 2) [55]. The particle size was found to have an inverse relationship with zeta potential. It is known that the zeta potential increases with the reduction in particle size [57].

$$Y_3 = 9.60 + 2.24X_1 - 0.79X_2 + 0.40X_3 - 1.04X_1X_2 - 1.34X_{12} - 1.5X_{22}. \tag{3.3}$$

## 3.5. Desirability and validation of the model

The desirability criteria and overlay plot of BBD, obtained from Design-Expert software, were helped to find the optimum combination of independent variables to formulate NPs (figure 4; electronic supplementary material, figure S5). The desirability score (figure 4) of the design was found to be 0.884, which implied precise outcome of the study. It was applied to reduce the variation in process and optimize the nanoformulation according to desirable properties. The optimized values of final formulation (TLM-PLGA-NPs) lay between the selected ranges mentioned in table 5. The optimization technique generated predicted results based on the predetermined values of the dependent variables. Hence the experimental values (actual) of TLM-PLGA-NPs were compared with the predicted values of dependent variables to confirm the validity of the optimized formulation (electronic supplementary material, figure S4). The experimental data regarding particle size, encapsulation efficiency and zeta potential of optimized formulation showed less than 10% difference as compared with the predicted response (table 5). The experimental value of drug loading of the nanoformulation was found to be

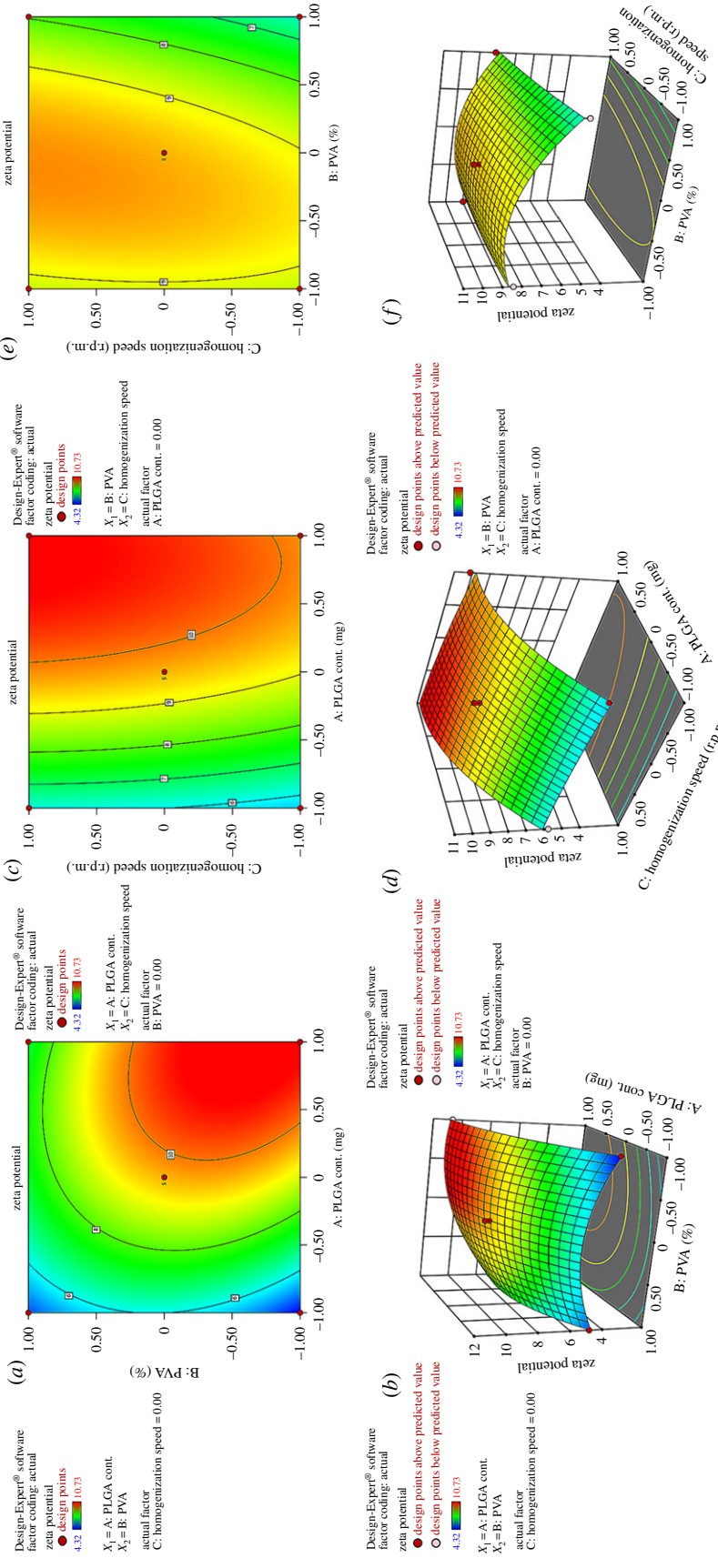

**Figure 3.** Contour plots and three-dimensional response surface plots showing the effect of different variables on the zeta potential of TLM-loaded NPs.

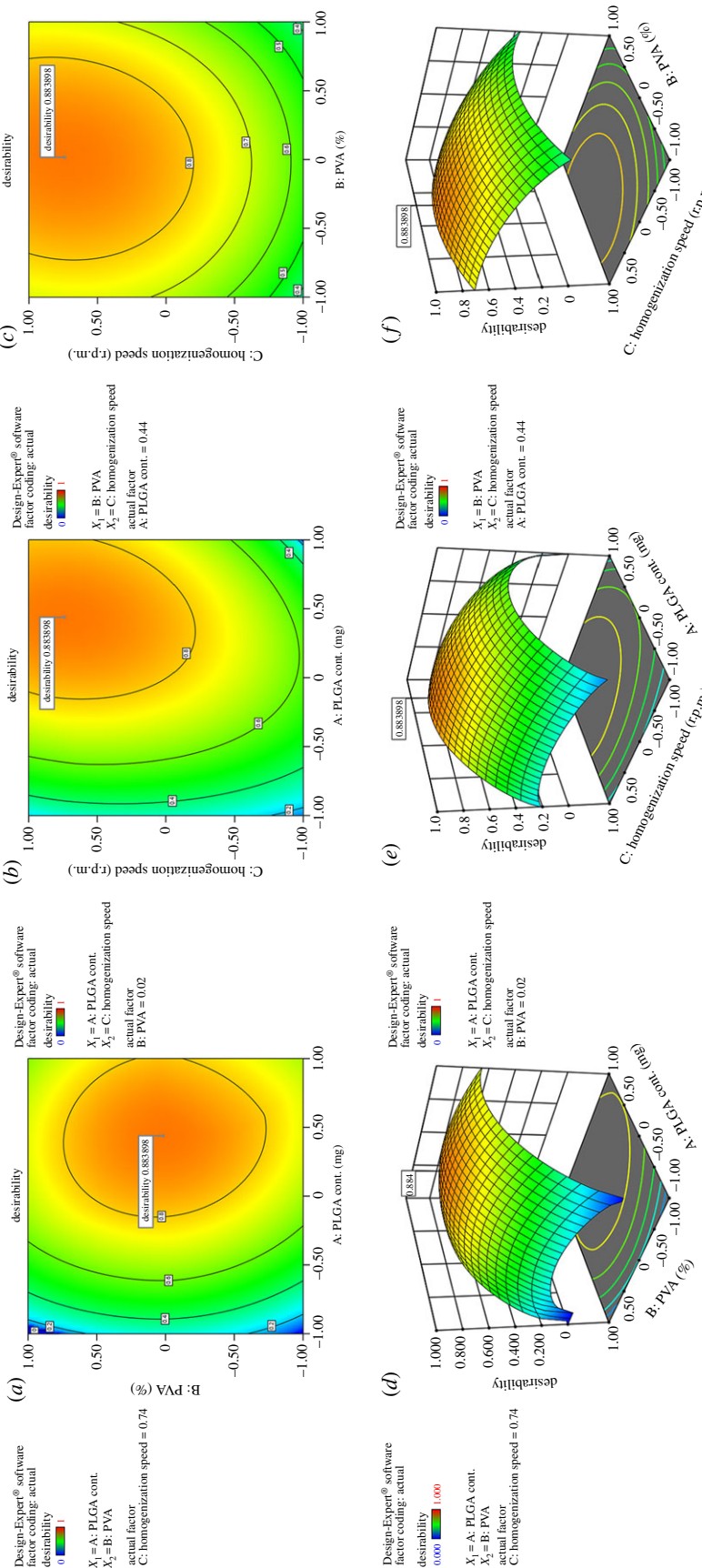

**Figure 4.** Contour and three-dimensional representation of optimization graph with desirability score.

**Table 5.** Comparison of the predicted and experimental values of the response variables of TLM-PLGA-NPs.

| name | goal | lower | upper | importance | predicted value | experimental value | bias (%) |
|---|---|---|---|---|---|---|---|
| PLGA cont. | is in range | −1 | 1 | 3 | 0.438 | | |
| PVA% | is in range | −1 | 1 | 3 | 0.015 | | |
| homogenization speed | is in range | −1 | 1 | 3 | 0.738 | | |
| particle size | minimize | 202.25 | 317.29 | 3 | 220.03 | 232.4 | −5.62 |
| encapsulation efficiency | maximize | 35.41 | 81.29 | 3 | 73.96 | 79.21 | −7.09 |
| zeta potential | maximize | 4.32 | 10.73 | 3 | 10.55 | 9.92 | 5.98 |

76.31%. The above values were calculated from final optimized formulation (TLM-PLGA-NPs) as suggested by Design-Expert software. The predicted and experimental data for optimization of TLM-PLGA-NPs are summarized in table 5.

## 3.6. Characterization of nanoparticles

### 3.6.1. FTIR study

The compatibility between drug and excipients was investigated in physical mixture and drug-loaded NPs by FTIR study. The FTIR spectra of TLM, PLGA, PVA, TLM-PLGA-PVA physical mixture and TLM-loaded NPs were measured in a range of 3600 to 400 cm$^{-1}$ and presented in figure 5. The characteristic peaks of TLM were detected at 3059.17 cm$^{-1}$ due to aromatic C–H stretching of benzene rings, 2960.44 cm$^{-1}$ due to aliphatic C–H bond stretching, 1697.23 cm$^{-1}$ due to the carbonyl group (C=O) of carboxylic acid and 1456.97 cm$^{-1}$ due to C=C aromatic group. The presence of the peak at 749.85 cm$^{-1}$ in the fingerprint region may be due to the ring vibration of 1,2-disubstituted benzene. The spectra of PLGA showed characteristic peaks at 1758.28 cm$^{-1}$ (C=O stretching of carbonyl group), 1455.40 cm$^{-1}$ (C–H stretching from methyl group), 1189.29 and 1090.94 cm$^{-1}$ (C–O stretching vibration) [58]. The figure (figure 5e) showed low intense principal peaks (3423.68 cm$^{-1}$ for O–H stretching from inter and intra molecular hydrogen bonds, 2939.18 cm$^{-1}$ for stretching of C–H from alkyl groups) of PVA which confirmed successful removal of surfactant from NPs. The previously described typical peaks of TLM were also observed in physical mixture and NPs. There were no major shifts in the characteristic peaks of drug and excipients which indicated that there was no evidence of significant chemical interaction between them. The result confirmed the chemical integrity of the dispersed drug into polymeric NPs.

### 3.6.2. DSC and XRD study

The physical property of both API and polymer have an influence on the therapeutic application of the drug [59]. In the present study, the content of the drug and polymer was varied in different ratios to form a statistical model. As a result the drug may entrap in different quantities within the polymeric NPs, which can affect the release profile of the formulation. Therefore, the drug may coexist across the polymer in different quantities within the NPs, which can affect the release profile of the formulation [19]. Further, during formulation, drug and polymer were introduced with different solvents and surfactants. Therefore, to identify any unwanted interaction between drug and polymer, regarding the development of formulation, DSC, XRD and LC-MS/MS analysis were performed. The thermal graph of DSC analysis was compared among the polymer, drug, TLM-PLGA physical mixtures and optimized TLM-PLGA-NPs (figure 6). The thermogram of figure 6a demonstrates an endothermic peak at 268.31°C corresponds to the melting temperature of TLM that confirmed the crystalline structure of the drug. No characteristic melting peak of TLM was found in the thermogram profile obtained from TLM-PLGA-NPs (figure 6d). In this thermogram, the absence of phase transitions relative to TLM justified the amorphous state of the drug. Further, PLGA showed glass transition temperature ($T_g$) at 56.56°C, which is the characteristic of its amorphous nature. The thermogram of

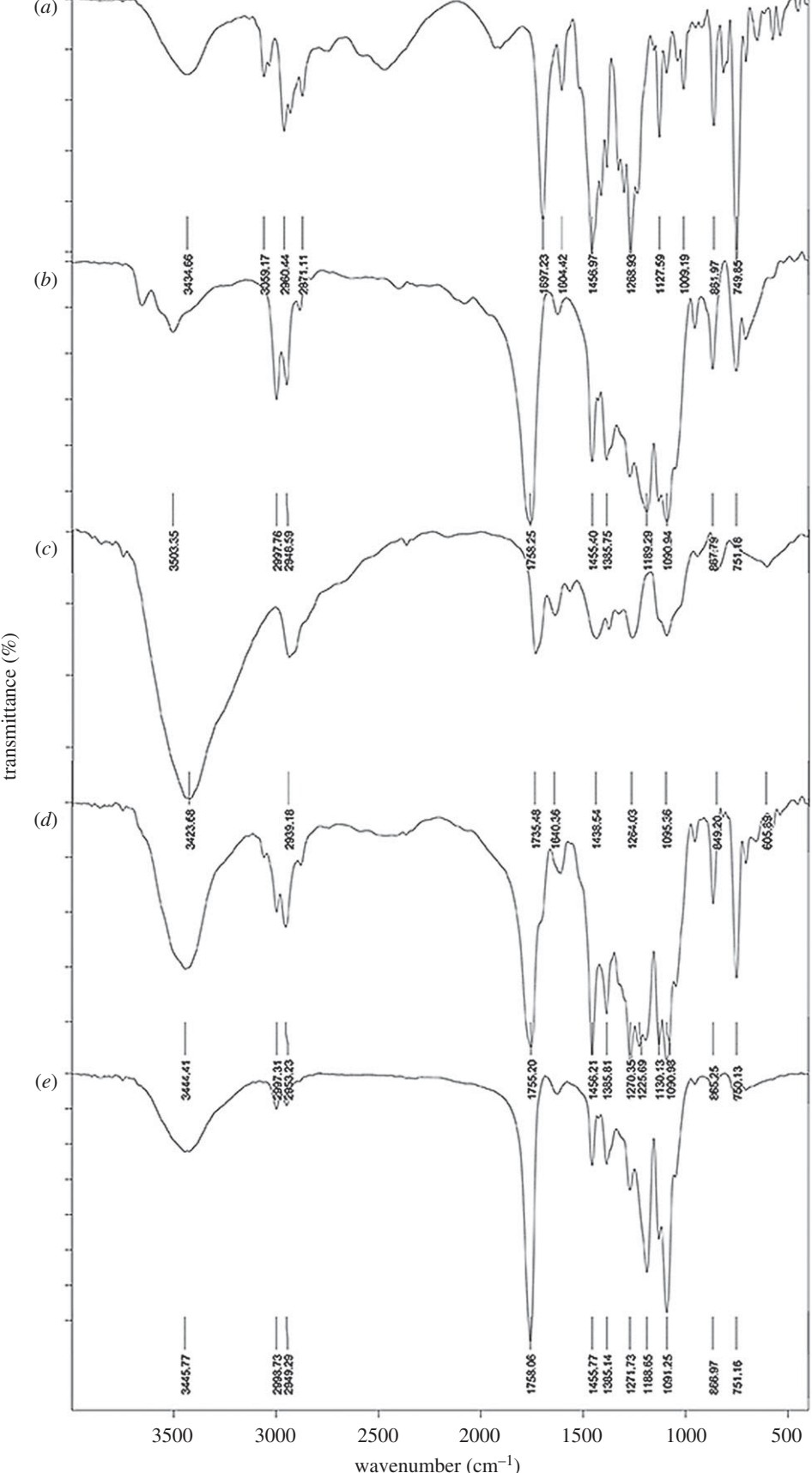

**Figure 5.** Fourier-transform infrared (FTIR) spectra of (*a*) TLM, (*b*) poly(D,L-lactic-co-glycolide) (PLGA), (*c*) polyvinyl alcohol, (*d*) physical mixture and (*e*) optimized formulation TLM-PLGA-NPs-02.

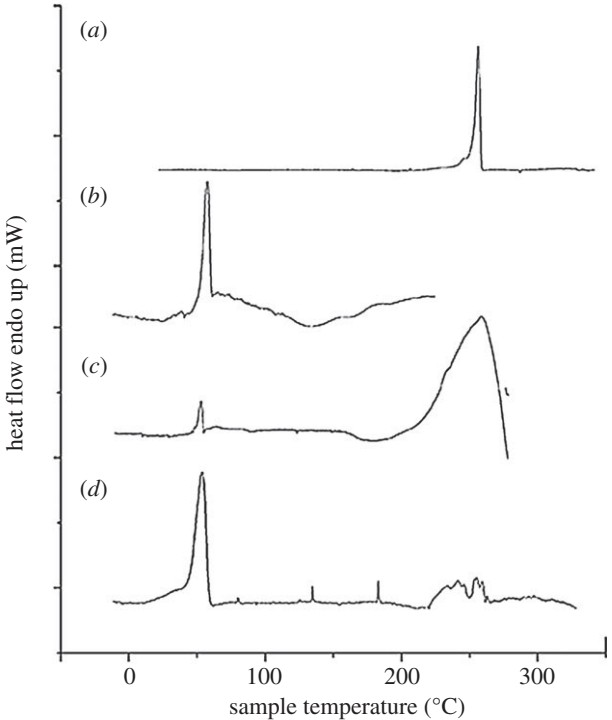

**Figure 6.** DSC thermograms of (*a*) TLM, (*b*) PLGA, (*c*) physical mixture and (*d*) TLM-PLGA-NPs-02.

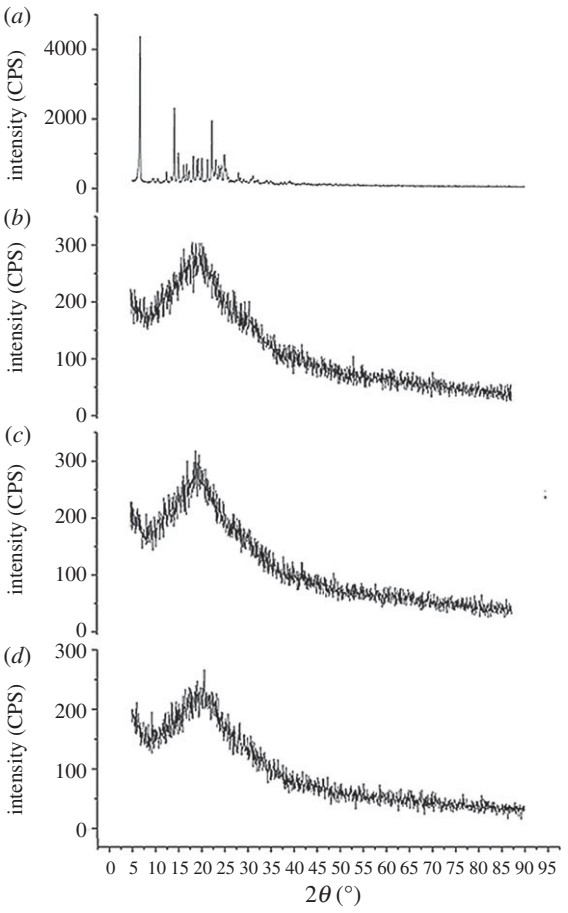

**Figure 7.** XRD studies of (*a*) TLM, (*b*) PLGA, (*c*) blank NPs and (*d*) TLM-PLGA-NPs-02.

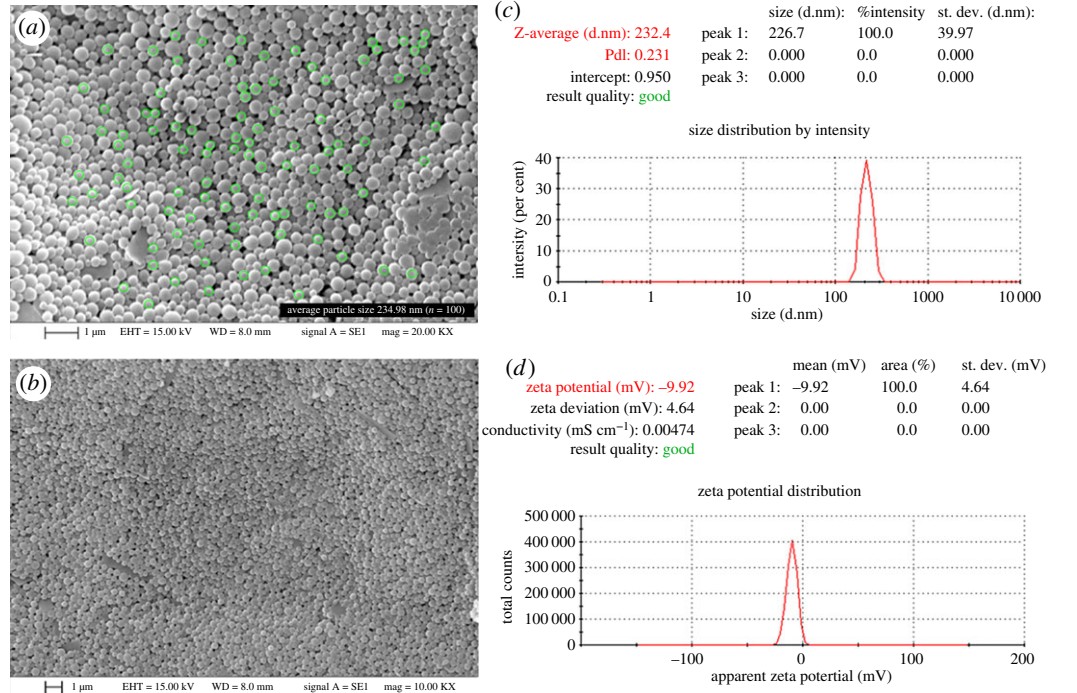

**Figure 8.** SEM images of TLM-PLGA-NPs-02 (*a*) magnification at 20KX (*b*) magnification at 10KX and (*c*) particle size distributions and (*d*) zeta potential.

formulated NPs exhibited only a peak at 56.56°C ($T_g$ of PLGA) which indicated no interaction occurred between TLM and PLGA during the manufacturing process. The decreased enthalpy ($\Delta H$) in thermogram data also represented a reduction in crystallinity of TLM for the TLM-PLGA-NPs.

The physical state like crystallinity of the compounds was also determined by XRD analysis since it may affect the release characteristic of the drug from NPs [60]. The diffractogram of pure TLM showed diffraction peaks at 6.7°, 14.2°, 15.0° and 22.3° in figure 7*a*, which indicated the crystalline nature of TLM. Among them, the typical peaks at 6.7°, 14.2° and 22.3° were particularly distinctive. On the other hand, the diffractogram of TLM-PLGA mixture exhibited less intense diffraction peak (figure 7*d*) of TLM as compared with pure drug. A more regular baseline verifies the amorphous nature of PLGA shown in figure 7*b*. Eventually the disappearance of intense peaks at a same diffraction angle in TLM-PLGA-NPs indicated that TLM may have undergone a physical transformation to amorphous solid dispersion.

Further TLM was scanned and quantified in LC-MS/MS before and after preparation of NPs. The +Q1 and MS2 scan confirmed the total mass (M + H) of TLM and its daughter ion respectively. The scan results (transition pair) of standard and extracted samples of TLM were found to be identical, i.e. m/z 515.3–276.2. The quantification of drug was performed in the multi-reaction monitoring mode of LC-MS/MS, where the intensity of peaks regarding NPs showed approximately similar response with respect to the reference standard of TLM. The +Q1 scan also demonstrated similarities in the results of the compound in both pure and extracted samples (data not shown). This indicated that TLM may not have any interaction with the excipients.

### 3.6.3. Evaluation of surface morphology, particle size and zeta potential

The image obtained from SEM was analysed to obtain more information on the morphology, particle size and aggregation phenomenon of optimized NPs that are illustrated in figure 8*a*,*b*. SEM micrograph at different magnification levels showed that TLM-loaded NPs had nearly monodispersed spherical shape with no sign of aggregation. The smooth surface of NPs may contribute to the uniform sustained release of the drug. The particle size and zeta potential of the optimized NPs were shown in figure 8*c*,*d*. A detailed discussion about particle size and zeta potential has been given above.

### 3.7. *In vitro* drug release study

A sustained release profile is one of the main characteristics of polymeric NPs, as suggested by Makadia & Siegel [46]. In this study, PLGA was used to formulate polymeric NPs. PLGA degrades by

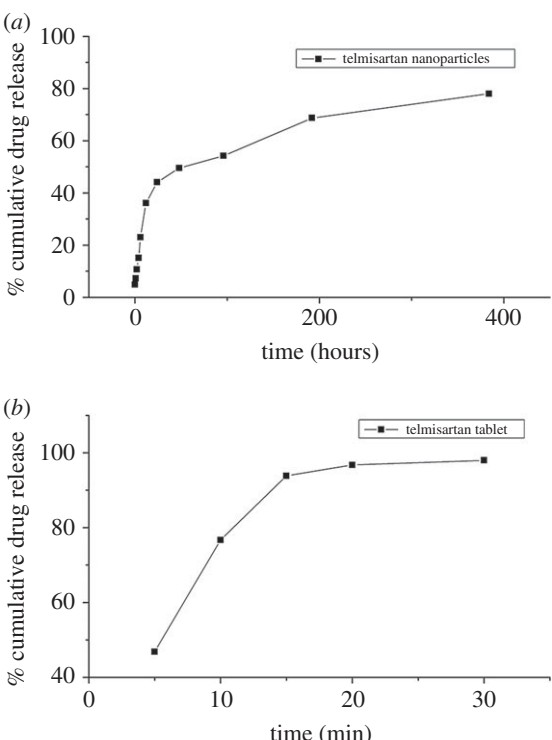

**Figure 9.** *In vitro* release profile of (*a*) TLM–PLGA-NPs and (*b*) TLM tablet.

**Table 6.** Results of curve fitting of the *in vitro* release of TLM from optimized formulation.

| model | $R^2$ | release exponent ($n$) |
|---|---|---|
| Korsmeyer–Peppas | 0.925 | 0.436 |
| Higuchi | 0.901 | — |
| zero order | 0.695 | — |
| first order | 0.430 | — |

hydrolysis of its ester linkages in the presence of water. According to the literature, PLGA (50 : 50) has a large degradation period of 102 days [61]. Consequently, a prolonged (384 hour) release profile was observed for the TLM-PLGA-NPs in phosphate buffer saline (pH 7.4). *In vitro* release study of TLM tablets (80 mg) and TLM-PLGA-NPs was performed in comparative perspective for confirmation of better therapeutic efficacy, as shown in figure 9. As compared with TLM tablet, TLM-PLGA-NPs showed a prolonged release profile.

Drug-encapsulated NPs showed biphasic release kinetic in phosphate buffer solution (pH 7.4). The pH of the medium was selected to mimic the human blood pH. Further, several kinetic models were employed to determine the possible release mechanism [33]. The curve fitting data of *in vitro* release kinetics are summarized in table 6, in accordance to the high regression coefficient. Korsmeyer–Peppas model ($R^2 = 0.925$) appeared to be best-fitted model contrary to the zero-order model, first-order model and Higuchi model. Release data of optimized formulation were fitted to the Korsmeyer–Peppas equation that proposed release of TLM from the TLM-PLGA-NPs followed Fickian diffusion ($n = 0.436$) [35]. These results indicate that the TLM release from polymeric NPs is through simple diffusion-controlled mechanism.

## 4. Conclusion

Based on the physiological determinants and intended use of the therapeutics, a pharmaceutical dosage form should be designed. Specific prerequisites must be established within the dosage form for uniform

delivery of the therapeutics across the biological barrier. In this regard, response surface methodology serves best to select and optimize process parameters to upgrade formulation technique of polymeric NPs with the help of Design-Expert software. The application of statistical design reduced the development time as well as human effort simultaneously with the economic production of accurate and precise NPs. The present study revealed that predefined dosage form (NPs) could be precisely prepared with proper implementation of statistical models. In this regard major independent parameters like PLGA cont., PVA% and homogenization speed were identified statistically by comparing the coefficient of determination ($R^2$) depending on the intended outcome of polymeric drug delivery system. The process flow expanded with the screening of preliminary factors and optimized them statistically by analysing its relative impact on responses by PBD and BBD respectively. The selected parameters were solely associated with applied double emulsion solvent evaporation technique which was further optimized by BBD for reduction of the experimental run. The derived polynomial equations and three-dimensional plots also aided in predicting the quantitative effect of the independent parameters on the response variables. Additionally, the validation of the model ensured successful implementation of the study in the development of modern drug delivery system. Based on the above model the TLM-loaded polymeric NPs can be formulated according to the desirability of diverse application perspective. The preparative method of polymeric NPs becomes more convincing with a highly impressive drug loading percentage. The study provides an opportunity to engineer the formulation pathway and navigate the design space in manufacturing perspective of polymeric NPs. In conclusion, the scientific investigation adopting implementation of statistical models can be an effective solution for manufacturing hurdles, which can improve overall quality of the products at a reasonable cost.

Data accessibility. All the data, figures and tables supporting this article have been included into the manuscript. Electronic supplementary material is available online at https://dx.doi.org/10.6084/m9.figshare.c.4582628.

Authors' contributions. P.S., S.B. and T.K.P. designed the study. P.S. and S.B. performed the experiments and characterization testing. P.S., S.B. and T.K.P. interpreted the results and wrote the manuscript. All authors gave final approval for publication.

Competing interests. We have no competing interests.

Funding. The above study was performed under the financial support of 'UGC-BSR Research Fellowship in Science for Meritorious Students 2012–2013', awarded to P.S. in Basic Scientific Research (UGC-BSR) scheme by University Grand Commission (UGC), India.

Acknowledgements. The authors gratefully acknowledge University Grand Commission (UGC) for providing financial support to conduct the study. The authors are thankful to Aurobindo Pharma, Hyderabad, India for providing TLM as a gift sample. The authors are also grateful to the Bioequivalence Study Centre, Jadavpur University, Kolkata, India for providing essential facilities and instruments and Centre for Research in Nanoscience and Nanotechnology, Calcutta University, Kolkata, India for providing facilities to carry out scanning electron microscopy (SEM). We would like to express our special thanks of gratitude to my teacher (Dr S. G. Dastider) for her moral support. We would also like to thank Mr Arnab De for providing all the necessary information for developing the study.

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
