## [Reviewer comments · Royal Society Open Science]

Review History

RSOS-182215.R0 (Original submission)

Review form: Reviewer 1

Is the manuscript scientifically sound in its present form?

No

Are the interpretations and conclusions justified by the results?

Yes

Is the language acceptable?

No

Is it clear how to access all supporting data?

No

Do you have any ethical concerns with this paper?

No

Have you any concerns about statistical analyses in this paper?

No

Recommendation?

Reject

Comments to the Author(s)

1. English proofreading is recommended.
2. The Design-Expert ver.7 used in the manuscript is a trial version, which is a clear violation of the trial software agreement. A valid software purchase should be made for academic research.
3. Telmisartan has a very good bioavailability p.o., and thanks to the high protein binding, the elimination half-life is over 24h, i.e. the release profile of telmisartan tablet is figure 8B is not relevant and misleading from a pharmacokinetic perspective, as the tablet is to design for instant release for good bioavailability. In Fig. 8A, the telmisartan nanoparticles released near 40% in just 24 h, and release another 40% till 400h. Unless the nanoparticle is giving a blood circulation time way over 400h (pharmacokinetic data not shown), the rest of the 60% telmisartan are unlikely to maintain a stable blood drug concentration within the therapeutic window (as the release speed is not consistent. The short blood circulation implying by the large >200 nm particle size also implies that ~50% of the unreleased drug will not be bioavailable before the nanoparticle is eliminated. If the maintenance of the blood drug concentration at the steady state requires multiple injections, then the invention claimed in the manuscript is not valid anymore. In that case, the author should design a multi-release kinetic packages/particles to keep the drug concentration at steady state.
4. The author used PVA without correlating the full name poly-vinyl alcohol at the same place. In the DOE the author used both polymer conc. and PVA and factors yet forgetting that PVA is also a polymer. Such terminology is causing confusion. "Polymer conc." should be changed into PLGA etc.
5. The experimental design, raw data, and analysis are missing for information listed in table 1. Discussion and justifications are needed for the selection/elimination of the process parameters.
6. The information in the ANOVA table is incomplete, the reviewer was forced to redo the DOE analysis in Design-Expert version 11 to acquire the information, e.g. lack of fit and pure error etc.
7. All DOE figures are directly copy-pasted from Design-Expert, additional efforts should be made to make sure the readability of the text on the graph.
8. The author mentioned the optimization of the engineering to maximize the EE%, surface charges, while minimizing the particle sizes, yet the optimization data/results are completely missing from the manuscript. How the author calculated the optimization to obtain Table 5 is missing. The optimization graph with disability score should also be shown. The optimized parameters were also missing. The reviewer was forced to run the analysis on Design-Expert to discover such information. The reader of the manuscript should not have to do so.
9. The characterization of the materials was not very informative. The FT-IR spectra should also include functional group identification. DSC thermograms are not conclusive and lacking descriptions and discussion. The XRD data is not informative at all. The particle sizes from the SEM should be analyzed and compared with the hydrodynamic sizes obtained by DLS.

10. All figures should include captions. There's only figure names at the moment.

Review form: Reviewer 2

Is the manuscript scientifically sound in its present form?

Yes

Are the interpretations and conclusions justified by the results?

Yes

Is the language acceptable?

Yes

Is it clear how to access all supporting data?

Yes

Do you have any ethical concerns with this paper?

No

Have you any concerns about statistical analyses in this paper?

No

Recommendation?

Accept with minor revision (please list in comments)

Comments to the Author(s)

This manuscript describes optimization of parameters for preparation of a formulation containing drug-loaded polymeric nanoparticles (NPs) using statistical design. Telmisartan (TLM), an anti-hypertension drug and commercial PLGA polymer are used along with PVA as surfactant. The process controlling variables are identified by applying Plackett-Burman design and critical process parameters are obtained using Box-Behnken design. Effect of independent variable parameters such as polymer concentration, surfactant concentration, homogenization speed on particle size, zeta potential and encapsulation efficiency of the TLM-loaded polymeric NPs was studied using response surface plots. The experimental results match well with those predicted by statistical design using the software. Various models including, Korsmeyer-Peppas and Higuchi are applied to study the kinetics of drug release. The NPs are well characterized using FT-IR, DSC, SEM and XRD techniques. The manuscript studies the effect of chemical nature and physical state of the drug on formation of stable and uniform NPs.

Overall, the study has been systematically carried out using scientifically appropriate methodologies. The use of statistical design for the formulation of TLM-loaded polymeric NPs is novel. The results of the manuscript would be of interest to researchers working in the area of drug formulation using polymeric nanocarriers. The manuscript is recommended for publication in Royal Society Open Science after certain concerns as given below are addressed.

- 1) Formulation number of TLM-PLGA-NPs should be mentioned in figure captions.
- 2) Conclusion section needs revision. It should specifically summarize the results of this study.
- 3) Please provide full name for the abbreviation TLM-PLGA in the summary (section 1).

Review form: Reviewer 3

Is the manuscript scientifically sound in its present form?

Yes

Are the interpretations and conclusions justified by the results?

Yes

Is the language acceptable?

Yes

Is it clear how to access all supporting data?

Yes

Do you have any ethical concerns with this paper?

No

Have you any concerns about statistical analyses in this paper?

Yes

Recommendation?

Major revision is needed (please make suggestions in comments)

Comments to the Author(s)

These authors studied the application of statistical design to evaluate critical process parameters and optimize formulation technique of polymeric nanoparticles. Their results are interesting to the reader. I have the following questions: (1) As their models showed, the interaction and quadratic effects are significant. So I suggest that these authors do the cubic effect investigation, which might have higher interaction among the factors. (2) in the optimization conditions, what are the dependent parameters?

Decision letter (RSOS-182215.R0)

30-Apr-2019

Dear Mr Sarkar:

Manuscript ID: RSOS-182215

Title: "Application of statistical design to evaluate critical process parameters and optimize formulation technique of polymeric nanoparticles."

Thank you for submitting the above manuscript to Royal Society Open Science. Your paper was sent to reviewers and their comments are included at the bottom of this letter. I apologise that this took longer than usual.

In view of the concerns raised by the reviewers, the manuscript has been rejected in its current form. However, a new manuscript may be submitted which takes into consideration these comments.

Please note that resubmitting your manuscript does not guarantee eventual acceptance, and that your resubmission will be subject to peer review before a decision is made.

Your resubmitted manuscript should be submitted by 28-Oct-2019. If you are unable to submit by this date please contact the Editorial Office.

On behalf of the Subject Editor Professor Anthony Stace and the Associate Editor Professor Kim Jelfs

REVIEWER(S) REPORTS:
Associate Editor Comments to Author ():
RSC Associate Editor:
Comments to the Author:
(There are no comments.)

RSC Subject Editor:
Comments to the Author:
(There are no comments.)

Reviewers' Comments to Author:
Reviewer: 1

Comments to the Author(s)

1. English proofreading is recommended.
2. The Design-Expert ver.7 used in the manuscript is a trial version, which is a clear violation of the trial software agreement. A valid software purchase should be made for academic research.

3. Telmisartan has a very good bioavailability p.o., and thanks to the high protein binding, the elimination half-life is over 24h, i.e. the release profile of telmisartan tablet is figure 8B is not relevant and misleading from a pharmacokinetic perspective, as the tablet is to design for instant release for good bioavailability. In Fig. 8A, the telmisartan nanoparticles released near 40% in just 24 h, and release another 40% till 400h. Unless the nanoparticle is giving a blood circulation time way over 400h (pharmacokinetic data not shown), the rest of the 60% telmisartan are unlikely to maintain a stable blood drug concentration within the therapeutic window (as the release speed is not consistent. The short blood circulation implying by the large >200 nm particle size also implies that ~50% of the unreleased drug will not be bioavailable before the nanoparticle is eliminated. If the maintenance of the blood drug concentration at the steady state requires multiple injections, then the invention claimed in the manuscript is not valid anymore. In that case, the author should design a multi-release kinetic packages/particles to keep the drug concentration at steady state.

4. The author used PVA without correlating the full name poly-vinyl alcohol at the same place. In the DOE the author used both polymer conc. and PVA and factors yet forgetting that PVA is also a polymer. Such terminology is causing confusion. "Polymer conc." should be changed into PLGA etc.

5. The experimental design, raw data, and analysis are missing for information listed in table 1. Discussion and justifications are needed for the selection/elimination of the process parameters.

6. The information in the ANOVA table is incomplete, the reviewer was forced to redo the DOE analysis in Design-Expert version 11 to acquire the information, e.g. lack of fit and pure error etc.

7. All DOE figures are directly copy-pasted from Design-Expert, additional efforts should be made to make sure the readability of the text on the graph.

8. The author mentioned the optimization of the engineering to maximize the EE%, surface charges, while minimizing the particle sizes, yet the optimization data/results are completely missing from the manuscript. How the author calculated the optimization to obtain Table 5 is missing. The optimization graph with disability score should also be shown. The optimized parameters were also missing. The reviewer was forced to run the analysis on Design-Expert to discover such information. The reader of the manuscript should not have to do so.

9. The characterization of the materials was not very informative. The FT-IR spectra should also include functional group identification. DSC thermograms are not conclusive and lacking descriptions and discussion. The XRD data is not informative at all. The particle sizes from the SEM should be analyzed and compared with the hydrodynamic sizes obtained by DLS.

10. All figures should include captions. There's only figure names at the moment.

Reviewer: 2

Comments to the Author(s)

This manuscript describes optimization of parameters for preparation of a formulation containing drug-loaded polymeric nanoparticles (NPs) using statistical design. Telmisartan (TLM), an anti-hypertension drug and commercial PLGA polymer are used along with PVA as surfactant. The process controlling variables are identified by applying Plackett-Burman design and critical process parameters are obtained using Box-Behnken design. Effect of independent variable parameters such as polymer concentration, surfactant concentration, homogenization speed on particle size, zeta potential and encapsulation efficiency of the TLM-loaded polymeric NPs was studied using response surface plots. The experimental results match well with those predicted by statistical design using the software. Various models including, Korsmeyer-Peppas and Higuchi are applied to study the kinetics of drug release. The NPs are well characterized using FT-IR, DSC, SEM and XRD techniques. The manuscript studies the effect of chemical nature and physical state of the drug on formation of stable and uniform NPs.

Overall, the study has been systematically carried out using scientifically appropriate methodologies. The use of statistical design for the formulation of TLM-loaded polymeric NPs is novel. The results of the manuscript would be of interest to researchers working in the area of drug formulation using polymeric nanocarriers. The manuscript is recommended for publication in Royal Society Open Science after certain concerns as given below are addressed.

- 1) Formulation number of TLM-PLGA-NPs should be mentioned in figure captions.
- 2) Conclusion section needs revision. It should specifically summarize the results of this study.
- 3) Please provide full name for the abbreviation TLM-PLGA in the summary (section 1).

Reviewer: 3

Comments to the Author(s)

These authors studied the application of statistical design to evaluate critical process parameters and optimize formulation technique of polymeric nanoparticles. Their results are interesting to the reader. I have the following questions: (1) As their models showed, the interaction and quadratic effects are significant. So I suggest that these authors do the cubic effect investigation, which might have higher interaction among the factors. (2) in the optimization conditions, what are the dependent parameters?

Author's Response to Decision Letter for (RSOS-182215.R0)

See Appendix A.

RSOS-190896.R0

Review form: Reviewer 3

Is the manuscript scientifically sound in its present form?

Yes

Are the interpretations and conclusions justified by the results?

Yes

Is the language acceptable?

Yes

Is it clear how to access all supporting data?

Yes

Do you have any ethical concerns with this paper?

No

Have you any concerns about statistical analyses in this paper?

No

Recommendation?

Accept with minor revision (please list in comments)

Comments to the Author(s)

This authors well did their research. There are small issues, such as : (1) State-ease Inc should be Stat-ease Inc. (2) I suggest that these authors released all the data as the supportive information with the manuscript. I suggest accepting this manuscript for publication with revision.

Decision letter (RSOS-190896.R0)

12-Jun-2019

Dear Mr Sarkar:

Title: Application of statistical design to evaluate critical process parameters and optimize formulation technique of polymeric nanoparticles.

Manuscript ID: RSOS-190896

Thank you for submitting the above manuscript to Royal Society Open Science. On behalf of the Editors and the Royal Society of Chemistry, I am pleased to inform you that your manuscript will be accepted for publication in Royal Society Open Science subject to minor revision in accordance with the referee suggestions. Please find the reviewers' comments at the end of this email.

The reviewers and handling editors have recommended publication, but also suggest some minor revisions to your manuscript. Therefore, I invite you to respond to the comments and revise your manuscript.

Because the schedule for publication is very tight, it is a condition of publication that you submit the revised version of your manuscript before 21-Jun-2019. Please note that the revision deadline will expire at 00.00am on this date. If you do not think you will be able to meet this date please let me know immediately.

- 1) A text file of the manuscript (tex, txt, rtf, docx or doc), references, tables (including captions) and figure captions. Do not upload a PDF as your "Main Document".
- 2) A separate electronic file of each figure (EPS or print-quality PDF preferred (either format should be produced directly from original creation package), or original software format)

- 3) Included a 100 word media summary of your paper when requested at submission. Please ensure you have entered correct contact details (email, institution and telephone) in your user account
- 4) Included the raw data to support the claims made in your paper. You can either include your data as electronic supplementary material or upload to a repository and include the relevant doi within your manuscript
- 5) All supplementary materials accompanying an accepted article will be treated as in their final form. Note that the Royal Society will neither edit nor typeset supplementary material and it will be hosted as provided. Please ensure that the supplementary material includes the paper details where possible (authors, article title, journal name).

Best wishes,

Dr Laura Smith
Publishing Editor, Journals

On behalf of the Subject Editor Professor Anthony Stace and the Associate Editor Professor Kim Jelfs.

RSC Associate Editor
Comments to the Author:
(There are no comments.)

Reviewer comments to Author:
Reviewer: 3

Comments to the Author(s)

This authors well did their research. There are small issues, such as : (1) State-ease Inc should be Stat-ease Inc. (2) I suggest that these authors released all the data as the supportive information with the manuscript. I suggest accepting this manuscript for publication with revision.

Author's Response to Decision Letter for (RSOS-190896.R0)

See Appendix B.

Decision letter (RSOS-190896.R1)

24-Jun-2019

Dear Mr Sarkar:

Title: Application of statistical design to evaluate critical process parameters and optimize formulation technique of polymeric nanoparticles.

Manuscript ID: RSOS-190896.R1

It is a pleasure to accept your manuscript in its current form for publication in Royal Society Open Science. The chemistry content of Royal Society Open Science is published in collaboration with the Royal Society of Chemistry.

On behalf of the Subject Editor Professor Anthony Stace and the Associate Editor Professor Kim Jelfs.

RSC Associate Editor
Comments to the Author:
(There are no comments.)

Reviewer(s)' Comments to Author:

Appendix A

Response to reviewer:

	Manuscript ID: RSOS-182215	Title: "Application of statistical design to evaluate critical process parameters and optimize formulation technique of polymeric nanoparticles."
Sl. No.	Reviewers' Comments to Author:	Response to the reviewer.
Reviewer: 1		
1	English proofreading is recommended.	The manuscript was further grammar and spell checked with the help of Ginger software.
2	The Design-Expert ver.7 used in the manuscript is a trial version, which is a clear violation of the trial software agreement. A valid software purchase should be made for academic research.	As suggested by the respective reviewer, the licensed copy of Design-Expert ver.11 software had purchased and employed to re-evaluate the entire study statistically.
3	Telmisartan has a very good bioavailability p.o., and thanks to the high protein binding, the elimination half-life is over 24h, i.e. the release profile of telmisartan tablet is figure 8B is not relevant and misleading from a pharmacokinetic perspective, as the tablet is to design for instant release for good bioavailability.	I completely agree with the point of view represented by the respective reviewer, regarding bioavailability of TLM as first line treatment of hypertension, but I afraid that I have gone through several published literature where in vitro release of TLM was exhibited release study as an immediate release tablet (As suggested by Reviewer) which is similar to my work depicted in Figure 9B (Previously 8B) (80% of TLM was released within 20 minutes). The clinical study of TLM on a large number of patients, showed a huge variation in pharmacokinetics (C _{max} -297.47 ±282.36 ng/ml, CV% 94.92; T _{1/2} -30.41±21.58 h, CV% 70.85; N=57) [London, 21 July 2011 EMA/777686/2011, Committee for Medicinal Products for Human Use (CHMP), European Medicines Agency, 2011]. As TLM belongs to the family of BCS class 2, it represents poor water solubility that leads them towards variable bioavailability. Therefore, at a time all the subjects, under the treatment of TLM cannot achieve the uniform therapeutic activity. Being a highly recommended antihypertensive therapeutics, the exposure of the drug in systemic circulation needs to be very accurate towards the control of hypertension. In this context, therapeutic drug monitoring may seem to be a solution regarding the accurate dose determination, but may fail in the practical field of expense and wide availability.
	In Fig. 8A, the telmisartan nanoparticles released	Based on the physical nature and pharmacokinetics variations, TLM (BCS Class 2) was selected as a model drug, subjected to be encapsulated within polymeric NPs to overcome the oral

	near 40% in just 24 h, and release another 40% till 400h. Unless the nanoparticle is giving a blood circulation time way over 400h (pharmacokinetic data not shown), the rest of the 60% telmisartan are unlikely to maintain a stable blood drug concentration within the therapeutic window (as the release speed is not consistent. The short blood circulation implying by the large >200 nm particle size also implies that ~50% of the unreleased drug will not be bioavailable before the nanoparticle is eliminated. If the maintenance of the blood drug concentration at the steady state requires multiple injections, then the invention claimed in the manuscript is not valid anymore. In that case, the author should design a multi-release kinetic packages/particles to keep the drug concentration at steady state.	drug delivery associated difficulties. Moreover, in this study, we had tried to develop a precise formulation technique in nanoscale dimension by identifying major parameters influencing developmental process where particle size was projected in 200-240nm with maximum encapsulation efficiency and zeta potential. The typical size range was chosen based on the literature review, where the range was found most effective for intended oral application of optimized NPs (1-4). We have particularly focused on the formulation technique and selection of product or process influencing parameters. Eventually, the characterization of optimized formulation became an important part of the study for ensuring the accuracy and quality of the product. Therefore, we have performed the in vitro release kinetics in phosphate buffer solution at pH 7.4. From the in vitro release study of NPs, it can be concluded that in vitro release kinetics follows Korsmeyer–Peppas model which indicates the release of polymeric dosage form and it followed Fickian diffusion. Therefore, in oral administration solubility of the drug will not be a rate limiting parameter to achieve their desired concentration in the systemic circulation. Additionally, improvement of in vivo pharmacokinetics regarding the developed formulation is not an objective of this study.
4	The author used PVA without correlating the full name poly-vinyl alcohol at the same place. In the DOE the author used both polymer conc. and PVA and factors yet	Thank you for your valuable suggestion. The full name of PVA was mentioned at the very beginning along with the abbreviated form accordingly. Further polymer conc. was changed into PLGA conc. in the figures and tables. Additionally, in the manuscript, PLGA was mentioned as polymer (Summery -1) and PVA was indicated as surfactant (section 3.2).

	forgetting that PVA is also a polymer. Such terminology is causing confusion. “Polymer conc.” should be changed into PLGA etc.	
5	The experimental design, raw data, and analysis are missing for information listed in table 1. Discussion and justifications are needed for the selection/elimination of the process parameters.	As directed, Table 1 was edited with the detailed information regarding Plackett–Burman design. In addition, a brief justification on selection/elimination of process parameters was also included in the manuscript under section 4.1.
6	The information in the ANOVA table is incomplete, the reviewer was forced to redo the DOE analysis in Design-Expert version 11 to acquire the information, e.g. lack of fit and pure error etc.	The entire statistical analysis was re-investigated in a licensed version of Design Expert 11 software and a complete ANOVA table was incorporated instead of previous one.
7	All DOE figures are directly copy-pasted from Design-Expert, additional efforts should be made to make sure the readability of the text on the graph.	The high resolution images regarding the design of experiment were incorporated within the manuscript.
8	The author mentioned the optimization of the engineering to maximize the EE%, surface charges, while minimizing the particle sizes, yet the optimization data/results are completely missing from the manuscript. How the author calculated the optimization to obtain Table 5 is missing. The optimization graph with	As directed, the optimization part was also accomplished during re-evaluation of the study, using license version Design-Expert software. All the data associated with the engineering of EE%, surface charges and particle sizes were included in the manuscript. Additionally, as mentioned, the calculation part, optimization graph (Fig. 4) with a desirability score were also added and refurbished in Table 5.

	disability score should also be shown. The optimized parameters were also missing. The reviewer was forced to run the analysis on Design-Expert to discover such information. The reader of the manuscript should not have to do so.	
9	The characterization of the materials was not very informative. The FT-IR spectra should also include functional group identification. DSC thermograms are not conclusive and lacking descriptions and discussion. The XRD data is not informative at all. The particle sizes from the SEM should be analyzed and compared with the hydrodynamic sizes obtained by DLS.	The detailed information regarding FT-IR, DSC, and XRD has been included and discussed in the “ Results and Discussion ” section with appropriate references. Further, the particle size of NPs from corresponding SEM images was calculated and compared with the hydrodynamic sizes obtained by DLS. In this context, a comparative view was also represented in Fig.8 (Previously Fig. 7).
10	All figures should include captions. There’s only figure names at the moment.	I understand your concern regarding figure captions. All the figure and table captions were enlisted under “ Figure and table captions ” according to journal specifications.
Reviewer: 2		
1	Formulation number of TLM-PLGA-NPs should be mentioned in figure captions.	As recommended by the reviewer, the optimized NPs (TLM-PLGA-NPs) name and number was mentioned in figure captions.
2	Conclusion section needs revision. It should specifically summarize the results of this study.	As suggested by the respective reviewer, the conclusion section was revised and included in the manuscript.
3	Please provide full name for the abbreviation TLM-PLGA in the summary (section 1).	In the manuscript, full name, Telmisartan loaded PLGA nanoparticles have been incorporated for the abbreviation TLM-PLGA in the summary (section 1).

Reviewer: 3		
1	As their models showed, the interaction and quadratic effects are significant. So I suggest that these authors do the cubic effect investigation, which might have higher interaction among the factors.	In fit summary part, Design-Expert software suggested that the cubic model is aliased. Therefore the quadric effects were considered for processing by the software itself.
2	In the optimization conditions, what are the dependent parameters?	The study was navigated towards the achievement of desired particle size, encapsulation efficiency and zeta potential. Further, in the optimization condition the dependent parameters was set just same based on the desired output of particle size (minimum), zeta potential (maximum) and encapsulation efficiency (maximum) of the nanoformulation.

References.

1. Lundy DJ, Chen K-H, Toh EKW, Hsieh PCH. Distribution of Systemically Administered Nanoparticles Reveals a Size-Dependent Effect Immediately following Cardiac Ischaemia-Reperfusion Injury. *Scientific Reports*. 2016;6:25613.
2. Alexis F, Pridgen E, Molnar LK, Farokhzad OC. Factors affecting the clearance and biodistribution of polymeric nanoparticles. *Molecular pharmaceutics*. 2008;5(4):505-15.
3. Wang H, Li Q, Reyes S, Zhang J, Xie L, Melendez V, et al. Formulation and particle size reduction improve bioavailability of poorly water-soluble compounds with antimalarial activity. *Malaria research and treatment*. 2013;2013:769234.
4. Kulkarni SA, Feng SS. Effects of particle size and surface modification on cellular uptake and biodistribution of polymeric nanoparticles for drug delivery. *Pharmaceutical research*. 2013;30(10):2512-22.

Appendix B

Response to reviewer:

	Manuscript ID: RSOS-190896	Title: "Application of statistical design to evaluate critical process parameters and optimize formulation technique of polymeric nanoparticles."
Sl. No.	Reviewers' Comments to Author:	Response to the reviewer.
Reviewer: 3		
1	State-ease Inc should be Stat-ease Inc.	As suggested by the respective reviewer State-ease Inc was corrected to Stat-ease Inc.
2	I suggest that these authors released all the data as the supportive information with the manuscript.	The authors provided all the supporting data as "electronic supplementary material" along with the manuscript.